# The Effect of Eating Speed on Sarcopenia, Obesity, and Sarcopenic Obesity in Older Adults: A 16-Year Cohort Study Using the Korean Genome and Epidemiology Study (KoGES) Data

**DOI:** 10.3390/nu17060992

**Published:** 2025-03-12

**Authors:** Sae Rom Lee, Sang Yeoup Lee, Young Hye Cho, Youngin Lee, Jung In Choi, Ryuk Jun Kwon, Soo Min Son, Jeong Gyu Lee, Yu Hyeon Yi, Young Jin Tak, Seung Hun Lee, Gyu Lee Kim, Young Jin Ra, Eun Ju Park

**Affiliations:** 1Research Institute for Convergence of Biomedical Science and Technology, Pusan National University Yangsan Hospital, Yangsan 50612, Republic of Korea; sweetpea85@naver.com (S.R.L.); saylee@pnu.edu (S.Y.L.); younghye82@naver.com (Y.H.C.); ylee23@gmail.com (Y.L.); s1jungin@hanmail.net (J.I.C.); brain6@hanmail.net (R.J.K.); soo890624@naver.com (S.M.S.); 2Department of Family Medicine, Pusan National University Yangsan Hospital, Yangsan 50612, Republic of Korea; 3Department of Family Medicine, The School of Medicine, Pusan National University, Yangsan 50612, Republic of Korea; eltidine@hanmail.net (J.G.L.); eeugus@hanmail.net (Y.H.Y.); 03141998@hanmail.net (Y.J.T.); greatseunghun@daum.net (S.H.L.); happygaru@hanmail.net (G.L.K.); yjra80@naver.com (Y.J.R.); 4Department of Medical Education, Integrated Research Institute for Natural Ingredients and Functional Food, School of Medicine, Pusan National University, Yangsan 50612, Republic of Korea; 5Department of Family Medicine, Pusan National University Hospital, Busan 46241, Republic of Korea

**Keywords:** eating speed, older, sarcopenia, obesity, cohort comparison

## Abstract

**Background:** Sarcopenia and obesity are age-related conditions associated with dietary habits. However, the relationship between eating speed and sarcopenia, particularly in older adults >65 years of age remains unclear. **Methods:** To investigate the effect of eating speed on the occurrence of sarcopenia, obesity, and sarcopenic obesity, we examined eating speed, socioeconomic factors, and disease history of 6202 patients at baseline and their changes over a 16-year follow-up period. **Results:** A fast eating speed was significantly associated with a higher risk of developing sarcopenia (normal eating: HR 1.284, 95% CI 1.107–1.490; slow eating: HR 1.583, 95% CI 1.279–1.958). Slower eating was associated with a reduced risk of obesity (normal eating: HR 0.865, 95% CI 0.786–0.952; slow eating: HR 0.680, 95% CI 0.577–0.802). These trends were consistent among participants aged <65 years. Among participants aged ≥65 years, fast eating was associated with a higher incidence of sarcopenia (HR 1.603, 95% CI 1.119–2.298), but no significant relationship existed with obesity (normal eating: HR 0.846, 95% CI 0.623–1.150; slow eating: HR 0.792, 95% CI 0.537–1.168). **Conclusions:** Slow eating speed decreased the incidence of obesity but increased that of sarcopenia in adults aged <65 years. However, in adults aged ≥65 years, slow eating speeds increased the incidence of sarcopenia but did not reduce the incidence of obesity.

## 1. Introduction

Age-related changes include increased adipose tissue and decreased skeletal muscle mass. Sarcopenia is defined as the decline in muscle mass, muscle strength, and muscle function with age [1]. In assessing sarcopenia, muscle mass is measured, and either appendicular lean mass or skeletal muscle mass divided by height squared [2,3] or body mass index (BMI) [4,5] is used. Sarcopenia causes various adverse health effects, including fractures, functional decline, and increased mortality [6]. Obesity is associated with diabetes, hypertension, hyperlipidemia, coronary artery disease, certain cancers, and reduced life expectancy [7,8,9]. Obesity, characterized by abnormal fat accumulation, is commonly classified using BMI, though this may not fully capture fat distribution. Several studies have reported that abdominal circumference can better predict fat distribution and related diseases [10]. Low muscle mass and function combined with excessive adiposity is known as sarcopenic obesity, which often occurs in older adults [11,12,13]. Sarcopenic obesity is caused by various unhealthy lifestyles and physical factors related to aging [14,15,16].

Both sarcopenia and obesity are closely associated with lifestyle factors, particularly healthy eating habits. To date, studies have primarily focused on the role of dietary nutrients and caloric intake in sarcopenia and obesity. Additionally, many studies have shown that faster eating speed is associated with increased obesity and weight gain. Previous research has established a link between faster eating speed and obesity, though studies specifically addressing sarcopenia are scarce [17,18]. Moreover, research on the effects of eating speed on obesity, sarcopenia, and sarcopenic obesity across different age groups is lacking. Therefore, this study aimed to investigate the incidence of sarcopenia, obesity, and sarcopenic obesity in the Korean population according to eating speed using large-scale cohort data from the Korean Genome and Epidemiology Study (KoGES).

## 2. Methods

### 2.1. Study Design

The data used in this study were obtained from the KoGES cohort. This study involved 10,030 residents of the Ansan (urban)–Ansung (rural) region aged 40–69 years, with follow-up observations every 2 years between 2001 and 2016. Participants in the KoGES cohort were recruited voluntary participants who registered by telephone call, on-site, mail letters, or followed a media campaign. Participants included in the cohort were invited to a national medical university hospital or health checkup center and underwent examination. The methodological overview, design, and rationale for KoGES have been detailed previously [19]. From the initial screening of 10,030 participants, 6541 people remained after excluding those whose eating speeds were not surveyed. We further excluded 56 individuals with missing bioelectrical impedance analysis (BIA) records or waist circumference measurements. In addition, 283 participants were excluded owing to their drinking history, education level, marital status, income, social history of smoking, or the presence of comorbidities not being recorded. Consequently, 6202 participants were included in the final analysis. Participation in this study was voluntary, and informed consent was obtained from all participants. Ethical approval was obtained from the appropriate ethics committee board. This study was approved by the Ethics Committee of Pusan National University Yangsan Hospital (approval number: 05-2023-084).

### 2.2. Covariates

Demographic characteristics of the study population (age, sex, marital status, smoking status, alcohol consumption, medical history, daily physical activity, and disease history) were gathered through self-administered questionnaires created by KoGES. Alcohol intake was categorized into non-drinkers (those who never consumed alcohol), former drinkers (those who abstained at the time of the survey), and current drinkers (those who consumed alcohol at the time of the survey). Total physical activity duration and intensity were calculated to assess daily physical activity levels. Diet was confirmed using the semi-quantitative food frequency questionnaire (FFQ) developed by KoGES, a survey containing 103 food items. Participants were also queried about various diseases (diabetes mellitus, kidney disease, cerebrovascular disease, arthritis, congestive heart disease, coronary artery disease, chronic obstructive pulmonary disease, and cancer) that could potentially influence sarcopenia risk. In addition, socioeconomic factors such as income level and educational attainment were considered as covariates, as they can significantly impact dietary habits, healthcare access, and overall health outcomes [1]. Previous research has shown that individuals with a higher socioeconomic status (SES) are more likely to have balanced diets, better protein intake, and greater access to healthcare, all of which may mitigate sarcopenia risk [2]. Conversely, a lower SES has been linked to increased food insecurity, lower protein consumption, and higher rates of malnutrition, which could exacerbate the effects of slow eating on sarcopenia [3].

### 2.3. Definition of Meal Speed

Eating speed was assessed through a self-administered KoGES questionnaire, where participants indicated their eating speed. Those who took less than 10 min to finish a meal were categorized as having a ‘fast’ eating speed, those who took 10 to 20 min were categorized as having a ‘normal’ eating speed, and those who took 30 min or more were categorized as having a ‘slow’ eating speed. Based on these 2001 baseline survey responses, participants were divided into three groups as follows: fast, normal, and slow eating speeds.

### 2.4. Definition of Sarcopenia, Obesity, and Sarcopenic Obesity

Muscle mass was assessed using multifrequency BIA (InBody 3.0; Biospace, Seoul, Republic of Korea). Appendicular lean mass was measured and adjusted for height squared to derive the appendicular lean mass index (ALMI) in kg/m^2^. Sarcopenia was diagnosed according to the operational definitions outlined by the Asian Working Group for Sarcopenia. Sarcopenia was defined based on appendicular lean mass index (ALMI), ALMI < 7.0 kg/m^2^ in men and < 5.7 kg/m^2^ in women [20], while obesity was determined by waist circumference (WC), WC ≥ 90 cm in men, and ≥85 cm in women [21]. Sarcopenic obesity was diagnosed when both conditions coexisted. The occurrence of sarcopenia, obesity, and sarcopenic obesity was confirmed through tests conducted every two years from 2001 to 2016.

### 2.5. Statistical Analyses

Baseline characteristics are presented as mean ± standard deviation for continuous variables and as numbers (percentage) for categorical variables across meal speeds (fast, normal, and slow). Continuous variables were analyzed using *t*-tests, whereas categorical variables were assessed using chi-square tests.

Hazard ratios (HRs) were determined using Cox proportional hazards models to estimate the 95% confidence interval (CI) for the incidence of sarcopenia based on meal speed. In the Cox proportional hazards model, adjustments were made for age and sex in Model 1. In Model 2, daily protein intake (g) and total daily physical activity (measured in METs) were added. Additionally, Model 3 was adjusted for education level, marital status, income, alcohol consumption, smoking status, and the presence of comorbidities such as diabetes, kidney disease, stroke, degenerative or rheumatoid arthritis, congestive heart failure, coronary artery disease, and chronic obstructive pulmonary disease.

## 3. Results

Table 1 presents the characteristics of the study participants. The average age of the 6202 participants was 53.4 years, and 50.4% were men. Those who ate the slowest were the oldest and had the lowest WC and ASMI. Among slow eaters, protein intake was the lowest, and total physical activity was the highest. Participants with normal eating speed were more likely to have graduated from college or higher. The fast-eating group had the highest proportion of non-drinkers, while non-smokers were most common in the slow-eating group. Among those with slow eating speeds, a higher proportion had incomes exceeding 6 million won.

Table 2 describes the relationship between eating speed and occurrence of sarcopenia after adjusting for age, sex, METs, protein intake, education level, marital status, income, alcohol consumption, presence of comorbidities, and smoking status. This analysis was conducted for the total age group, the under-65 years group, and the over-65 years group. In both the total and under-65 age groups, slower eating speeds were associated with a higher incidence of sarcopenia compared to fast speed eating; this trend remained consistent after adjusting for all covariates in Model 3 (total group: normal eating speed HR [95% CI]: 1.284 [1.107–1.490], slow eating speed HR [95% CI]: 1.584 [1.279–1.958], under 65 years old group: normal-speed eating group HR [95% CI]: 1.366 [1.151–1.622], slow-speed eating group HR [95% CI]: 1.499 [1.145–1.962]). However, in individuals aged > 65 years, no difference was noted in the occurrence of sarcopenia between the normal eating speed (HR [95% CI]: 1.068 [0.796–1.432]) and slow eating speed groups. In contrast, the fast-eating group still had a high incidence of sarcopenia (HR [95% CI]: 1.603 [1.119–2.298]) compared to the slow-eating group.

Table 3 shows the relationship between eating speed and obesity after adjusting for age, sex, METs, protein intake, education level, marital status, income, alcohol consumption, presence of comorbidities, and smoking status. Slow eating speeds were associated with a lower incidence of obesity in participants overall and under 65 years (Model3, total group: HR [95% CI], normal: 0.865 [0.786–0.952], slow: 0.680 [0.577–0.802], under 65 years group: HR [95% CI], normal: 0.868 [0.784–0.961], slow: 0.651 [0.541–0.784]), but no such association was found in those over 65 (Model 3, HR [95% CI], normal: 0.846 [0.623–1.150], slow: 0.792 [0.537–1.168]).

Table 4 describes the relationship between eating speed and the incidence of sarcopenic obesity after adjusting for age, sex, METs, protein intake, education level, marital status, income, alcohol consumption, presence of comorbidities, and smoking status. No difference was observed in the occurrence of sarcopenic obesity according to eating speed across all the groups. 

## 4. Discussion 

In this study, we found that faster eating speeds were associated with a higher incidence of sarcopenia and sarcopenic obesity. In contrast, slow eating was associated with a lower incidence of obesity. Interestingly, in people aged over 65 years, slow eating speed increased the incidence of sarcopenia but did not reduce the incidence of obesity.

Several studies have investigated the relationship between eating speed, obesity, and metabolic syndrome. Previous research has shown that fast eating speeds are associated with obesity and weight gain. A meta-analysis conducted in 2015 reported results similar to those of the present study. In the 12 cross-sectional studies included in this meta-analysis, the group with a fast eating speed had a higher BMI (1.78 kg/m^2^ [95% CI, 1.53–2.04 kg/m^2^]) than the group with a slow eating speed. Additionally, three longitudinal studies confirmed that even after adjusting for various variables, obesity was more common in those with fast eating speeds [22]. Our study demonstrated consistent results, indicating that slower eating speeds were associated with lower rates of obesity than faster eating speeds.

However, studies on the relationship between eating speed and sarcopenia are limited. Only one cross-sectional [17] and one cohort [18] study conducted using KAMOGAWA-DM group data reported that slower eating speeds were associated with a higher incidence of sarcopenia in aged ≥65 years Type2DM patients. This result aligns with that of our study, which found that in the 65 years or older age group, the incidence of sarcopenia was higher in the slow-eating group than in the fast-eating group.

No studies have examined eating speed and the occurrence of obesity according to age groups. However, in a study conducted in China among participants aged 18–45 years, a correlation existed between faster eating speed, lower WC, and fat mass percentage. However, in the age group of 65 years or older, eating speed was not related to WC or fat mass percentage [23]. Our study also found that, among participants under the age of 65 years, slower eating speed was associated with a lower incidence of obesity; however, in those over 65 years, fast eating speed was not related to the occurrence of obesity. The body develops anabolic resistance in older adults [24]. In this study, fast eating in people over 65 years of age did not increase obesity but reduced sarcopenia. Through this study, it can be concluded that in the adult population over 65 years of age, slow eating speed causes sarcopenia and does not protect against obesity. Slow eating had a worse effect on older adults than on younger age groups.

No previous studies have examined the relationship between eating speed and sarcopenic obesity, and our study is the first to explore this relationship. In this study, there was no significant relationship between eating speed and incidence of sarcopenic obesity. However, accurate results were not obtained in the present study because the incidence of sarcopenic obesity was very low. Additional research using patient groups with a higher incidence of sarcopenic obesity is needed.

Previous studies have shown that fast eating is associated with obesity, NAFLD, and type 2 diabetes [14,25,26,27]. In contrast, research has shown that slow eating increases the risk of malnutrition [28]. While rapid eating speed increases the incidence of obesity, the pathological mechanism that reduces the occurrence of sarcopenia has not been accurately identified. However, studies have shown that a fast eating speed leads to high energy intake, whereas a slow eating speed results in low energy intake [29,30]. A recent study confirmed that obese participants chewed less frequently and ate faster than lean participants when consuming 1 g of food. In this study, slow eating speeds were associated with low energy intake, low postprandial ghrelin concentration, and high postprandial GLP-1 and cholecystokinin concentrations [31]. In another study, the PYY and GLP-1 concentrations increased slowly when meals were consumed [32]. However, this reduced energy intake may have negative consequences for muscle health, particularly in older adults who already experience anabolic resistance. Ghrelin stimulates growth hormone (GH) secretion and promotes the regulation of hunger and obesity through GH-independent mechanisms. Ghrelin encodes acylated ghrelin (AG) and unacylated ghrelin (UnAG) [33]. AG promotes hunger and obesity and GH secretion, which produces anti-inflammatory responses. UnAG mimics the anti-atrophic action of AG on skeletal muscles [34]. Since slow eating has been associated with lower ghrelin levels, it is possible that reduced AG and UnAG levels may contribute to muscle loss by decreasing GH secretion and its muscle-protective effects [35]. In addition to the role of hormones such as ghrelin and GLP-1, the regulation of eating speed may also be influenced by other physiological mechanisms, including the nervous system. The nervous system plays a crucial role in regulating eating behaviors through signals from the brain, particularly through the vagus nerve and central nervous system (CNS) mechanisms. These neural pathways can affect satiety signals, such as those mediated by the hypothalamus, which controls hunger and energy intake. Recent research has shown that the vagus nerve plays a significant role in slowing down the eating process by regulating gastric motility and signaling satiety to the brain, which may help prevent overeating and contribute to better metabolic health [36]. Furthermore, the combination of lower energy intake, altered hormonal responses, and age-related anabolic resistance may create a compounding effect, making older adults more susceptible to sarcopenia. Given that adequate energy and protein intake is crucial for maintaining muscle mass, further research is needed to determine whether dietary modifications or supplementation strategies could mitigate sarcopenia risk in slow eaters [37,38,39].

This study had several limitations. First, because the tool used to measure eating speed was a self-reported questionnaire, this introduces the potential for recall bias and individual judgment differences, which can affect the accuracy of the data. Future studies should develop and utilize tools that can directly measure eating speed, such as video analysis or wearable devices that track eating behavior in real-time. This would allow for more accurate and reliable measurements of eating speed and provide a clearer understanding of its impact on health outcomes. Second, the long follow-up period of 16 years is a strength of this study, allowing for the observation of long-term trends. However, it also raises concerns about the consistency of dietary habits and lifestyle factors over time. Changes in these factors could influence the outcomes of this study, and future research should consider more frequent assessments of dietary habits and lifestyle changes to control for these temporal variations. Third, this study was conducted using a Korean cohort. As Koreans exhibit a high degree of ethnic homogeneity, this may not be fully representative of other ethnic groups, and we suggest exploring diverse populations to assess whether the relationship between eating speed and sarcopenia holds across different ethnic backgrounds. Besides that, our study sample was derived from the KoGES cohort, which, while substantial in size, may not be fully representative of the entire Korean older adult population. Participants were selected based on specific regions and voluntary health examination participation, which may introduce selection bias. Therefore, caution is needed when generalizing the findings to the broader population. Future studies using nationally representative samples are warranted. Additionally, in this study, muscle mass was assessed using ALMI, and obesity was determined based on waist circumference. While these measures are commonly used and widely accepted in the field, it is worth noting that additional methods such as muscle strength testing (e.g., handgrip strength) or body fat percentage assessments could have provided more detailed information regarding sarcopenia, obesity, and sarcopenic obesity. Incorporating these additional methods in future studies may enhance the comprehensiveness and accuracy of the findings, providing a more robust understanding of the relationship between eating speed, obesity, and sarcopenia. Nevertheless, this study has several strengths. This is the first study to identify the relationship between eating speed, obesity, sarcopenia, and sarcopenic obesity in a population. Additionally, it has the advantage of identifying causal relationships through 16 years of long-term cohort study. Finally, it had the advantage of being a large-scale population study, making the research highly reliable.

## 5. Conclusions

In this study, the results showed that slow eating decreased the incidence of obesity but increased the incidence of sarcopenia. However, in adults aged over 65 years, the effect of slow eating speeds on reducing obesity disappeared, whereas the effect on increasing sarcopenia remained. Clinical interventions are required to identify and prevent the risk of sarcopenia in older adults who eat slowly. Potential strategies may include nutritional interventions, such as ensuring adequate protein intake, physical activity promotion, and behavioral approaches, such as modifying chewing habits or meal pacing education. However, implementing such interventions in clinical practice may face challenges, including age-related physiological changes (e.g., dental health and digestive capacity), socioeconomic constraints, and the difficulty of modifying long-standing dietary habits in older adults. Further research is needed to develop practical and effective intervention strategies tailored to this population.

## Figures and Tables

**Table 1 nutrients-17-00992-t001:** Baseline characteristics of the participants according to eating speed.

	Fast (*n* = 2665)	Normal (*n* = 2859)	Slow (*n* = 678)	Total (*N* = 6202)	*p*
Age (years)	52.3 ± 8.8	53.6 ± 9.0	56.5 ± 9.1	53.4 ± 9.0	<0.001
Men, n (%)	1402 (52.6)	1441 (50.4)	282 (41.6)	3125 (50.4)	<0.001
ALMI (kg/m^2^)	7.4 ± 0.8	7.3 ± 0.8	7.1 ± 0.8	7.3 ± 0.8	<0.001
Waist circumference (cm)	84.2 ± 8.7	82.8 ± 8.6	81.5 ± 9.0	83.3 ± 8.8	<0.001
BMI (kg/m^2^)	25.0 ± 3.2	24.3 ± 3.0	23.5 ± 3.1	24.5 ± 3.2	<0.001
Protein intake (g)	66.9 ± 29.4	65.8 ± 30.4	65.4 ± 33.5	66.2 ± 30.3	0.391
METs (kcal/h)	10,530.9 ± 6539.7	11,155.1 ± 6726.5	11,762.8 ± 6683.3	10,953.3 ± 6683.2	<0.001
Education					
Elementary	924 (34.7)	1089 (38.1)	363 (53.5)	2376 (38.3)	<0.001
Middle	629 (23.6)	647 (22.6)	132 (19.5)	1408 (22.7)	
High	880 (33.0)	856 (29.9)	129 (19.0)	1865 (30.1)	
University	232 (8.7)	267 (9.3)	54 (8.0)	553 (8.9)	
Marriage					
Alone	37 (1.4)	42 (1.5)	8 (1.2)	87 (1.4)	
Marry	2400 (90.1)	2568 (89.8)	589 (86.9)	5557 (89.6)	
Divorce	228 (8.6)	249 (8.7)	81 (11.9)	558 (9.0)	
Alcohol					
Current	1111 (41.7)	1274 (44.6)	360 (53.1)	2745 (44.3)	<0.001
Former	185 (6.9)	187 (6.5)	40 (5.9)	412 (6.6)	
None	1369 (51.4)	1398 (48.9)	278 (41.0)	3045 (49.1)	
Smoking					
None	1447 (54.3)	1620 (56.7)	422 (62.2)	3489 (56.3)	<0.001
Former	457 (17.1)	499 (17.5)	90 (13.3)	1046 (16.9)	
Current	761 (28.6)	740 (25.9)	166 (24.5)	1667 (26.9)	
Income					
Under 1 million won	990 (37.1)	1200 (42.0)	381 (56.2)	2571 (41.5)	<0.001
Under 2 million won	1196 (44.9)	1220 (42.7)	229(33.8)	2645 (42.6)	
Under 3 million won	412 (15.5)	386 (13.5)	61 (9.0)	859 13.9)	
Under 6 million won	67 (2.5)	53 (1.9)	7 (1.0)	127 (2.0)	
Disease					
DM	173 (6.5)	200 (7.0)	47 (6.9)	420 (6.8)	0.680
CKD	68 (2.6)	88 (3.1)	20 (2.9)	176 (2.8)	0.563
CVD	27 (1.0)	35 (1.2)	17 (2.5)	79 (1.3)	0.002
OA	135 (5.1)	135 (4.7)	40 (5.9)	310 (5.0)	0.384
CHF	11 (0.4)	8 (0.3)	2 (0.3)	21 (0.3)	0.660
MI	32 (1.2)	31 (1.1)	12 (1.8)	75 (1.2)	0.246
CAD	26 (1.0)	19 (0.7)	7 (1.0)	52 (0.8)	0.894
COPD	20 (0.8)	16 (0.6)	12 (1.8)	48 (0.8)	<0.001

ALMI = appendicular lean mass index; BMI = body mass index; CAD = coronary artery disease; CHF = chronic heart failure; CKD = chronic kidney disease; COPD = chronic obstructive pulmonary disease; CVD = cerebrovascular disease; DM = diabetes mellitus; METs = metabolic equivalents; MI = myocardial infarction; OA = osteoarthritis. Data are presented as number (with percentage) of patients, mean ± standard deviation for parametric data. *p*-value was obtained by student’s test or by chi-square test.

**Table 2 nutrients-17-00992-t002:** Hazard ratio of sarcopenia according to eating speed.

Model	Fast HR (95% CI)	Normal HR (95% CI)	Slow HR (95% CI)
40–69 yeas			
Model 1	1	1.281 (1.107–1.482) **	1.567 (1.269–1.934) **
Model 2	1	1.289 (1.112–1.495) **	1.589 (1.285–1.966) **
Model 3	1	1.284 (1.107–1.490) **	1.583 (1.279–1.958) **
Under 65 years			
Model 1	1	1.351 (1.141–1.600) **	1.506 (1.154–1.964) *
Model 2	1	1.369 (1.154–1.625) **	1.524 (1.164–1.995) *
Model 3	1	1.366 (1.151–1.622) **	1.499 (1.145–1.962) *
Over 65 years			
Model 1	1	1.103 (0.829–1.469)	1.597 (1.120–2.277) *
Model 2	1	1.083 (0.808–1.452)	1.618 (1.133–2.312) *
Model 3	1	1.068 (0.796–1.432)	1.603 (1.119–2.298) *

CI = confidence interval; HR = hazard ratio. Data are expressed as hazard ratio (95%, confidence interval). * *p* < 0.001, ** *p* < 0.05.

**Table 3 nutrients-17-00992-t003:** Hazard ratio of obesity according to eating speed.

Model	Fast HR (95% CI)	Normal HR (95% CI)	Slow HR (95% CI)
40–69 yeas			
Model 1	1	0.859 (0.781–0.945) *	0.691 (0.588–0.813) **
Model 2	1	0.853 (0.774–0.939) **	0.687 (0.583–0.810) **
Model 3	1	0.865 (0.786–0.952) *	0.680 (0.577–0.802) **
Under 65 years			
Model 1	1	0.868 (0.785–0.959) *	0.676 (0.564–0.811) **
Model 2	1	0.858 (0.775–0.950) *	0.662 (0.550–0.796) **
Model 3	1	0.868 (0.784–0.961) *	0.651 (0.541–0.784) **
Over 65 years			
Model 1	1	0.829 (0.619–1.111)	0.743 (0.508–1.085)
Model 2	1	0.837 (0.619–1.131)	0.765 (0.522–1.119)
Model 3	1	0.846 (0.623–1.150)	0.792 (0.537–1.168)

CI = confidence interval; HR = hazard ratio. Data are expressed as hazard ratio (95%, confidence interval) * *p* < 0.001, ** *p* < 0.05.

**Table 4 nutrients-17-00992-t004:** Hazard ratio of sarcopenic obesity according to eating speed.

Model	Fast HR (95% CI)	Normal HR (95% CI)	Slow HR (95% CI)
40–69 yeas			
Model 1	1	1.079 (0.756–1.54)	1.132 (0.670–1.912)
Model 2	1	1.164 (0.811–1.671)	1.198 (0.707–2.029)
Model 3	1	1.177 (0.818–1.693)	1.125 (0.661–1.916)
Under 65 years			
Model 1	1	1.428 (0.929–2.195)	1.019 (0.485–2.142)
Model 2	1	1.539 (0.996–2.376)	1.086 (0.515–2.287)
Model 3	1	1.610 (0.104–2.495)	1.061 (0.503–2.241)
Over 65 years			
Model 1	1	0.542 (0.281–1.043)	1.100 (0.517–2.338)
Model 2	1	0.569 (0.288–1.125)	1.164 (0.544–2.491)
Model 3	1	0.517 (0.242–1.060)	0.904 (0.395–2.071)

CI = confidence interval; HR = hazard ratio. Data are expressed as hazard ratio (95%, confidence interval).

## Data Availability

Data are contained within the article.

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
