# Peer review of "The Effect of Eating Speed on Sarcopenia, Obesity, and Sarcopenic Obesity in Older Adults: A 16-Year Cohort Study Using the Korean Genome and Epidemiology Study (KoGES) Data"

_nutrients, 2025, doi:10.3390/nu17060992_

Round 1

Reviewer 1 Report

Comments and Suggestions for Authors

The study presents findings on the relationship between eating speed and sarcopenia, particularly in older adults over 65 years of age.

The work concludes that slow eating decreases obesity incidence but paradoxically increases sarcopenia incidence in older adults. This contradiction highlights the complexity of dietary habits and their impact on health, suggesting that interventions must be nuanced and tailored to the individual needs of older adults. The study sample of 6,202 patients is substantial; however, it is important to consider the demographics of the participants. Are they representative of the broader population of older adults? This could impact the generalizability of the findings.

The manuscript does not provide insights into the underlying mechanisms driving the association between eating speed and sarcopenia. Understanding how eating speed influences muscle health could help in developing targeted interventions.

The authors emphasize the need for clinical interventions to mitigate the risk of sarcopenia in older adults who eat slowly. However, the study does not elaborate on what these interventions might entail, nor does it discuss potential barriers to implementing them in clinical practice.

While the study examines socioeconomic factors, it does not delve deeply into how these factors may interact with eating speed and health outcomes. This could be an avenue for further research, as socioeconomic status often influences dietary habits and access to healthcare.

The 16-year follow-up period is a strength, but it also raises questions about the consistency of dietary habits over time. Changes in lifestyle and health status could affect the outcomes, and the study could benefit from controlling for these variables.

Author Response

Comments 1:

The work concludes that slow eating decreases obesity incidence but paradoxically increases sarcopenia incidence in older adults. This contradiction highlights the complexity of dietary habits and their impact on health, suggesting that interventions must be nuanced and tailored to the individual needs of older adults. The study sample of 6,202 patients is substantial; however, it is important to consider the demographics of the participants. Are they representative of the broader population of older adults? This could impact the generalizability of the findings.

Response 1:

Thank you for your insightful comment regarding the representativeness of our study sample. We acknowledge that the KoGES cohort, while substantial in size, may not fully represent the entire Korean older adult population due to potential selection bias related to regional and voluntary participation factors. To address this concern, we have added a statement in the Limitation section of the manuscript, highlighting the need for caution when generalizing the findings. Additionally, we have emphasized the necessity of future studies using nationally representative samples to validate our results. The revised text is as follows:

“Besides that, our study sample was derived from the KoGES cohort, which, while substantial in size, may not be fully representative of the entire Korean older adult population. Participants were selected based on (e.g., specific regions, voluntary health examination participation), which may introduce selection bias. Therefore, caution is needed when generalizing the findings to the broader population. Future studies using nationally representative samples are warranted.”

We appreciate your valuable feedback, which has helped improve the clarity and rigor of our study.

Comments2:

The manuscript does not provide insights into the underlying mechanisms driving the association between eating speed and sarcopenia. Understanding how eating speed influences muscle health could help in developing targeted interventions.

Response2:

Thank you for your valuable comment. We agree that understanding the underlying mechanisms behind the association between eating speed and sarcopenia is critical for developing targeted interventions. To address this, we have expanded the Discussion section to provide a more detailed explanation of potential mechanisms.

The revised section now includes the following discussion:

“However, this reduced energy intake may have negative consequences for muscle health, particularly in older adults who already experience anabolic resistance. Ghrelin stimulates growth hormone (GH) secretion and promotes the regulation of hunger and obesity through GH-independent mechanisms. Ghrelin encodes acylated ghrelin (AG) and unacylated ghrelin (UnAG) [33]. AG promotes hunger and obesity and GH secretion, which produces anti-inflammatory responses. UnAG mimics the anti-atrophic action of AG on skeletal muscles [34]. Since slow eating has been associated with lower ghrelin levels, it is possible that reduced AG and UnAG levels may contribute to muscle loss by decreasing GH secretion and its muscle-protective effects [35]. Furthermore, the combination of lower energy intake, altered hormonal responses, and age-related anabolic resistance may create a compounding effect, making older adults more susceptible to sarcopenia. Given that adequate energy and protein intake is crucial for maintaining muscle mass, further research is needed to determine whether dietary modifications or supplementation strategies could mitigate sarcopenia risk in slow eaters.”

We appreciate your suggestion, as it has helped us enhance the clarity and depth of our discussion.

Comments3:

The authors emphasize the need for clinical interventions to mitigate the risk of sarcopenia in older adults who eat slowly. However, the study does not elaborate on what these interventions might entail, nor does it discuss potential barriers to implementing them in clinical practice.

Response3:

Thank you for your valuable comment. We acknowledge that our initial conclusion lacked details on potential clinical interventions and the barriers to their implementation. To address this, we have revised the Conclusion section to include examples of possible interventions, such as nutritional, physical activity, and behavioral strategies, as well as potential challenges in applying them in clinical settings. The revised text is as follows:

“Clinical interventions are required to identify and prevent the risk of sarcopenia in older adults who eat slowly. Potential strategies may include nutritional interventions, such as ensuring adequate protein intake, physical activity promotion, and behavioral approaches, such as modifying chewing habits or meal pacing education. However, implementing such interventions in clinical practice may face challenges, including age-related physiological changes (e.g., dental health, digestive capacity), socioeconomic constraints, and the difficulty of modifying long-standing dietary habits in older adults. Further research is needed to develop practical and effective intervention strategies tailored to this population.”

We appreciate your insightful suggestion, which has helped strengthen our discussion on the practical implications of our findings.

Comments4:

While the study examines socioeconomic factors, it does not delve deeply into how these factors may interact with eating speed and health outcomes. This could be an avenue for further research, as socioeconomic status often influences dietary habits and access to healthcare.

Response4:

Thank you for your valuable suggestion regarding socioeconomic status (SES) and its potential interaction with eating speed and health outcomes. We recognize that SES plays a significant role in shaping dietary habits and access to healthcare, which may in turn influence the relationship between eating speed and sarcopenia risk.

To address this, we have expanded the Covariates section to clarify that income level and education were considered as covariates in our analysis. We have also added a discussion on how individuals with lower SES may have limited access to high-quality protein sources and healthcare, potentially exacerbating the effects of slow eating on sarcopenia. Conversely, those with higher SES may benefit from greater dietary diversity and healthcare support, which could mitigate the risks associated with slow eating.

The revised text now includes:

“ In addition, socioeconomic factors such as income level and educational attainment were considered as covariates, as they can significantly impact dietary habits, healthcare access, and overall health outcomes [1]. Previous research has shown that individuals with higher socioeconomic status (SES) are more likely to have balanced diets, better protein intake, and greater access to healthcare, all of which may mitigate sarcopenia risk [2]. Conversely, lower SES has been linked to increased food insecurity, lower protein consumption, and higher rates of malnutrition, which could exacerbate the effects of slow eating on sarcopenia [3].”

We appreciate your insightful feedback, as it has allowed us to strengthen our discussion of SES as an important contextual factor influencing our study findings.

Comments5:

The 16-year follow-up period is a strength, but it also raises questions about the consistency of dietary habits over time. Changes in lifestyle and health status could affect the outcomes, and the study could benefit from controlling for these variables.

Response5:

Thank you for your insightful comment regarding the 16-year follow-up period. We agree that long-term studies face challenges related to the consistency of dietary habits and lifestyle over time, and that changes in these factors could influence the outcomes.

We plan to highlight the limitations of the study in the manuscript, emphasizing that future research should explore the impact of lifestyle changes over time, and consider more frequent assessments of dietary habits and health status to better account for temporal variations. This would help strengthen the ability to draw more definitive conclusions about the long-term effects of eating speed on sarcopenia risk. The revised text is as follows:

"Second, the long follow-up period of 16 years is a strength of the study, allowing for the observation of long-term trends. However, it also raises concerns about the consistency of dietary habits and lifestyle factors over time. Changes in these factors could influence the outcomes of the study, and future research should consider more frequent assessments of dietary habits and lifestyle changes to control for these temporal variations."

We hope this clarifies our approach and appreciate your feedback, which has helped improve the discussion around the study's design and its limitations.

We sincerely appreciate the reviewers' valuable comments and suggestions, which have greatly contributed to improving our manuscript. We have carefully considered all the feedback and made the necessary revisions accordingly. We hope that the revised manuscript meets the expectations of the reviewers and the editorial board.

Thank you for your time and effort in reviewing our work. We look forward to your further feedback.

Reviewer 2 Report

Comments and Suggestions for Authors

The paper “Effect of Eating Speed on Sarcopenia, Obesity, and Sarcopenic Obesity in Older Adults: A 16-Year Cohort Study Using the Korean Genome and Epidemiology Study (KoGES) Data” aimed to investigate the effect of eating speed on the occurrence of sarcopenia, obesity, and sarcopenic obesity. The research content is quite interesting, but the data seems to be insufficient. Here are some specific issues.

Comments:

Q1. The repetition rate of the manuscript is too high. It is unacceptable in the absence of a justifiable reason.

Q2. This study used the data from the KoGES cohort of the Korean population. Due to the high ethnic homogeneity of the Korean population, it is suggested that the discussion section further elaborates on the limitations of extrapolating the findings to other ethnic populations and the need for future similar studies on different ethnic groups to improve the generalizability of the study conclusion.

Q3. Eating speed was assessed through a self-reported questionnaire, which is highly subjective and may be subject to recall bias and individual judgment differences.

Q4. Specific measures (such as ALMI, waist circumference, etc.) were used in the definition of sarcopenia, obesity, and sarcopenic obesity, respectively. Although these indicators are more commonly used in related fields, more measures could be considered for inclusion

Q5. Although the data are detailed, some graphs can be added to make the research results more intuitive and clear for readers to understand and compare.

Q6. When discussing the potential mechanism of the relationship between eating speed and disease, although the role of some hormones (such as ghrelin and GLP-1) is mentioned, it is not thorough and comprehensive. It is recommended to further review the latest literature to explore other possible physiological mechanisms (such as nervous system regulation and the influence of intestinal flora) to enrich the interpretation of the study results.

Q7. The effective data in this article are not sufficient. It is suggested to further supplement the relevant data as much as possible, or to fully analyze the existing data to enrich the content of this manuscript.

Author Response

Q1. The repetition rate of the manuscript is too high. It is unacceptable in the absence of a justifiable reason.

Response1:

Thank you for your valuable feedback. I have carefully reviewed the manuscript and made the following revisions to address the repetition concerns:

Introduction: I have streamlined the definitions of obesity and clarified the relationship between eating speed, obesity, and sarcopenia to reduce redundancy and improve clarity.

Obesity, characterized by abnormal fat accumulation, is commonly classified using BMI, though this may not fully capture fat distribution.”

Previous research has established a link between faster eating speed and obesity, though studies specifically addressing sarcopenia are scarce [17, 18].”

Methods: I removed redundant explanations regarding eating speed and the definitions of sarcopenia and obesity, ensuring that the definitions are consistent throughout the manuscript.

“Eating speed was assessed through a self-administered KoGES questionnaire, where participants indicated their eating speed.”

“Sarcopenia was defined based on appendicular lean mass index (ALMI), ALMI < 7.0 kg/m² in men and < 5.7 kg/m² in women [20], while obesity was determined by waist circumference (WC),WC ≥ 90 cm in men and ≥ 85 cm in women [21]. Sarcopenic obesity was diagnosed when both conditions coexisted.

Results: I have refined the discussion of the relationship between eating speed, sarcopenia, obesity, and sarcopenic obesity, aiming to eliminate repetition and enhance the overall clarity of the presentation.

“In both the total and under-65 age groups, slower eating speeds were associated with a higher incidence of sarcopenia compared to fast speed eating; this trend remained con-sistent after adjusting for all covariates in Model 3 (total group: normal eating speed HR [95% CI]: 1.284 [1.107-1.490], slow eating speed HR [95% CI]: 1.584 [1.279-1.958], under 65 yesrs old group: normal-speed eating group HR [95% CI]: 1.366 [1.151-1.622], slow-speed eating group HR [95% CI]: 1.499 [1.145-1.962]).”

“Slow eating speeds were associated with a lower incidence of obesity in participants total and under 65 years (Model3, total group: HR [95% CI], normal: 0.865 [0.786-0.952], slow: 0.680 [0.577-0.802], under 65 years group: HR [95% CI], normal: 0.868 [0.784-0.961], slow: 0.651 [0.541-0.784]), but no such association was found in those over 65 (Model 3,HR [95% CI], normal: 0.846 [0.623-1.150], slow: 0.792 [0.537-1.168]).”

I hope these revisions meet your expectations and contribute to a more concise and clearer presentation of the findings. Thank you again for your helpful suggestions.

Q2. This study used the data from the KoGES cohort of the Korean population. Due to the high ethnic homogeneity of the Korean population, it is suggested that the discussion section further elaborates on the limitations of extrapolating the findings to other ethnic populations and the need for future similar studies on different ethnic groups to improve the generalizability of the study conclusion.

Response 2:

Thank you for your valuable feedback. Based on your suggestion, I have made the following revisions:

Limitations Section: I have added a discussion on the potential limitations of extrapolating the findings to other ethnic groups due to the high ethnic homogeneity of the Korean cohort. This addresses the need for future studies involving diverse ethnic populations to improve the generalizability of the results.

“may not be fully representative of other ethnic groups, and we suggest exploring diverse populations to assess whether the relationship between eating speed and sarcopenia holds across different ethnic backgrounds.”

I hope these revisions meet your expectations and improve the clarity and scope of the study. Thank you once again for your insightful suggestions.

Q3. Eating speed was assessed through a self-reported questionnaire, which is highly subjective and may be subject to recall bias and individual judgment differences.

Response3;

Thank you for your valuable feedback. Based on your suggestion, I have revised the Limitations section to highlight the subjectivity of the self-reported questionnaire, as well as the potential for recall bias and individual judgment differences. I have acknowledged these factors as limitations that may affect the accuracy of the measurement of eating speed. I have also emphasized the need for future research using more objective methods to measure eating speed.

“This introduces the potential for recall bias and individual judgment differences, which can affect the accuracy of the data. or inaccuracies in reporting..”

I hope this revision adequately addresses your concerns. Thank you again for your thoughtful comments.

Q4. Specific measures (such as ALMI, waist circumference, etc.) were used in the definition of sarcopenia, obesity, and sarcopenic obesity, respectively. Although these indicators are more commonly used in related fields, more measures could be considered for inclusion

Response4:

Thank you for your suggestion regarding the inclusion of additional measures for defining sarcopenia, obesity, and sarcopenic obesity. I have acknowledged this in the Limitations section of the Discussion. While we utilized commonly accepted indicators like ALMI and waist circumference, I mentioned that incorporating additional methods, such as muscle strength testing or body fat percentage, could provide more comprehensive insights.I hope this revision addresses your concern and clarifies the limitations of the current study.

“In this study, muscle mass was assessed using ALMI, and obesity was determined based on waist circumference. While these measures are commonly used and widely accepted in the field, it is worth noting that additional methods such as muscle strength testing (e.g., handgrip strength) or body fat percentage assessments could have provided more detailed information regarding sarcopenia, obesity, and sarcopenic obesity. Incorporating these additional methods in future studies may enhance the comprehensiveness and accuracy of the findings, providing a more robust understanding of the relationship between eating speed, obesity, and sarcopenia.”

Q5. Although the data are detailed, some graphs can be added to make the research results more intuitive and clear for readers to understand and compare.

Response5:

Thank you for your insightful suggestion. To enhance the clarity and intuitiveness of our research findings, we have added a new figure (Figure 1) that visually represents the data. We believe this will help readers better understand and compare the results.

Please let us know if you have any further suggestions. We truly appreciate your time and effort in reviewing our work.

Q6. When discussing the potential mechanism of the relationship between eating speed and disease, although the role of some hormones (such as ghrelin and GLP-1) is mentioned, it is not thorough and comprehensive. It is recommended to further review the latest literature to explore other possible physiological mechanisms (such as nervous system regulation and the influence of intestinal flora) to enrich the interpretation of the study results.

Response 6:

Thank you for your valuable feedback. We appreciate your suggestion to further explore the potential physiological mechanisms underlying the relationship between eating speed and disease. In response to your comment, we have expanded the discussion to include additional mechanisms, specifically the regulation by the nervous system.

“In addition to the role of hormones such as ghrelin and GLP-1, the regulation of eating speed may also be influenced by other physiological mechanisms, including the nervous system and intestinal flora. The nervous system plays a crucial role in regulating eating behaviors through signals from the brain, particularly through the vagus nerve and central nervous system (CNS) mechanisms. These neural pathways can affect satiety signals, such as those mediated by the hypothalamus, which controls hunger and energy intake. Recent research has shown that the vagus nerve plays a significant role in slowing down the eating process by regulating gastric motility and signaling satiety to the brain, which may help prevent overeating and contribute to better metabolic health [36].”

Once again, thank you for your insightful suggestion. We believe that these revisions strengthen the manuscript and provide a more thorough discussion of the factors influencing eating speed and its relationship to disease.

Q7. The effective data in this article are not sufficient. It is suggested to further supplement the relevant data as much as possible, or to fully analyze the existing data to enrich the content of this manuscript.

Response 7.

Thank you for your valuable feedback. We appreciate your suggestion to supplement or further analyze the data. However, after careful consideration, we believe that the current dataset sufficiently supports the main findings of our study. Therefore, we would like to proceed with the manuscript in its current form. We hope that our explanations and interpretations adequately address the concerns raised.

Please let us know if there are any specific aspects that require further clarification. We truly appreciate your time and effort in reviewing our work.

We sincerely appreciate the reviewers' valuable comments and suggestions, which have greatly contributed to improving our manuscript. We have carefully considered all the feedback and made the necessary revisions accordingly. We hope that the revised manuscript meets the expectations of the reviewers and the editorial board.

Thank you for your time and effort in reviewing our work. We look forward to your further feedback.

Round 2

Reviewer 2 Report

Comments and Suggestions for Authors

The quality of the manuscript has been significantly improved, and I have no further comments.